# Reduced *Staphylococcus* Abundance Characterizes the Lesional Microbiome of Actinic Keratosis Patients after Field-Directed Therapies

Jan Kehrmann,[a] Fiona Koch,[b] Skrollan Zumdick,[a] Anna Höwner,[b] Lara Best,[a] Lars Masshöfer,[c] Sarah Scharfenberg,[b] Michael Zeschnigk,[c] Jürgen C. Becker,[d,e] Dirk Schadendorf,[b,e] Jan Buer,[a] Alexander Roesch[b,e]

[a]Institute of Medical Microbiology, University Hospital Essen, University of Duisburg-Essen, Essen, Germany
[b]Department of Dermatology, University Hospital Essen, University Duisburg-Essen and German Cancer Partner site Essen/Düsseldorf, Essen, Germany
[c]Institute for Human Genetics, University Hospital Essen, University of Duisburg-Essen, Essen, Germany
[d]Departments of Translational Skin Cancer Research and Dermatology, University Hospital Essen, Essen, Germany
[e]German Cancer Consortium (DKTK), Partner Site Essen/Düsseldorf and German Cancer Research Center (DKFZ), Heidelberg, Germany

**ABSTRACT**   Skin microbiome dysbiosis with a *Staphylococcus* overabundance is a feature of actinic keratosis (AK) and squamous skin carcinoma (SCC) patients. The impact of lesion-directed treatments for AK lesions such as diclofenac (DIC) and cold atmospheric plasma (CAP) on the lesional microbiome is not established. We studied 321 skin microbiome samples of 59 AK patients treated with DIC 3% gel versus CAP. Microbial DNA from skin swabs taken before start of treatment (week 0), at the end of the treatment period (week 24), and 3 months after end of treatment (week 36) was analyzed after sequencing the V3/V4 region of the *16S rRNA* gene. The relative abundance of *S. aureus* was scrutinized by a *tuf* gene specific TaqMan PCR assay. The total bacterial load and both, relative and absolute abundance of *Staphylococcus* genus was reduced upon both therapies at week 24 and 36 compared to week 0. Notably, the lesional microbiome of patients responding to CAP therapy at week 24 was characterized by an increased relative abundance of *Corynebacterium* genus compared to nonresponders. A higher relative abundance of *Staphylococcus aureus* at week 36 was a feature of patients classified as nonresponders for both treatments 12 weeks after therapy completion. The reduction of the *Staphylococcus* abundance after treatment of AK lesions and alterations linked to treatment response encourage further studies for investigation of the role of the skin microbiome for both, the carcinogenesis of epithelial skin cancer and its function as predictive therapeutic biomarker in AK.

**IMPORTANCE**   The relevance of the skin microbiome for development of actinic keratosis (AK), its progression into squamous skin cancer, and for field-directed treatment response is unknown. An overabundance of staphylococci characterizes the skin microbiome of AK lesions. In this study, analyses of the lesional microbiome from 321 samples of 59 AK patients treated with diclophenac gel versus cold atmospheric plasma (CAP) revealed a reduced total bacterial load and reduced relative and absolute *Staphylococcus* genus abundance upon both treatments. A higher relative *Corynebacterium* abundance was a feature of patients classified as responders at the end of CAP-treatment period (week 24) compared with nonresponders and the *Staphylococcus aureus* abundance of patients classified as responders 3 months after treatment completion was significantly lower than in nonresponders. The alterations of the skin microbiome upon AK treatment encourage further investigations for establishing its role for carcinogenesis and its function as predictive biomarker in AK.

**KEYWORDS**   skin microbiome, CAP, cold atmospheric plasma, diclofenac, actinic keratosis, treatment

Address correspondence to Jan Kehrmann, Jan.Kehrmann@uk-essen.de.
The authors declare no conflict of interest.

Actinic keratoses (AK) are premalignant keratotic lesions of the skin that may progress to squamous cell carcinoma (SCC), the second most common skin malignancy worldwide after basal cell carcinomas (1). Chronic UV-light exposure is a major risk factor for development of AK. Immunosuppressive medication, high age, light skin color, and chronic inflammation additionally predispose for AK development (2, 3). Microorganisms are supposed to contribute to chronic inflammatory skin diseases such as atopic dermatitis (4), but the relevance of the skin microbiome for AK development, for its progression into SCC, and for AK treatment response is not clarified yet. However, an increased prevalence of β-human papillomavirus and *Staphylococcus aureus* characterizes the skin microbiome of AK patients (2). β-human papillomavirus has been suggested to promote the survival of UV-light exposed DNA-damaged cells and thereby promote the evolvement of cancer cells (5). *Staphylococcus aureus* may cause or maintain skin inflammation and induces beta-defensin production of keratinocyte cell lines increasing keratinocyte proliferation (6). AK treatment includes field-directed anti-inflammatory therapies, e.g., diclofenac (DIC) 3% gel that inhibits prostaglandin synthesis (7). We have shown that cold atmospheric argon plasma (CAP), a partly ionized gas that contains reactive oxygen and nitrogen species, is a new option for AK treatment with comparable efficiency to conventional treatments and less side effects compared with DIC (8). In the present study, we analyzed the skin microbiome longitudinally in patients with AK treated for a period of 24 weeks with CAP versus DIC 3% gel of two comparable AK regions. To our knowledge, changes of the skin microbiome of AK patients in response to field-directed treatments have not been investigated longitudinally yet. We observed a reduction of the absolute and relative abundance of *Staphylococcus* after treatment, mainly attributed to coagulase-negative staphylococci (CoNS). Treatment response to either DIC or CAP therapy at week 36 was linked to a lower relative *S. aureus* abundance in responders than in nonresponding patients. The AK microbiome of patients responding to CAP therapy at the time point of therapy completion (week 24) was characterized by a higher relative abundance of the *Corynebacterium* genus compared to the AK microbiome of patients not responding to CAP treatment.

## RESULTS

**Regional differences in the skin microbiome of AK lesions.** The skin microbiome of two comparable AK areas was investigated in 59 participants. In each patient, one area was treated with CAP and the other with DIC. Skin microbiome swabs were taken at three time points: before start of treatment (week 0), at the end of treatment (week 24), and at the end of the study at week 36. The mean age of the individuals was 74 years and 56 of 59 patients (95%) were male. The skin microbiome from AK lesions was investigated from capillitium in 51, torso in 5, and extremities in 3 patients. As the skin microbiome of healthy individuals varies significantly depending on the body region, we first studied the extent of the differences of the skin microbiome linked to the AK localization. Alpha diversity metrics, including Shannon diversity, richness, and evenness were significantly higher in skin swabs from extremity AK lesions compared to those from capillitium and torso (Fig. 1A). In addition, Shannon diversity was higher from AK areas of capillitium compared to the torso region. Significant differences in the skin microbiome beta diversity were determined by Principal Coordinates analysis (PCoA) of Bray Curtis distance matrix with $P < 0.001$ (Fig. 1B). While the genus *Cutibacterium* (previous nomenclature *Propionibacterium)* and phylum Actinobacteria were enriched in the torso region and were biomarkers of the skin microbiome for this location as identified by LEfSe (Linear discriminant analysis Effect Size) analysis, the family *Neisseriaceae* was a biomarker for the skin microbiome of patients with AK of the capillitium (Fig. 1 C and D). Several taxa, including the genera *Streptococcus*, *Enhydrobacter,* and the phylum Proteobacteria were enriched and biomarkers for AK of extremities.

**Relative *Staphylococcus* abundance is reduced in the AK skin microbiome of treated patients.** We determined the differences in the skin microbiome of AK patients before start of therapy (week 0), at the time point of treatment completion (week 24), and 3 months after completion of treatment (week 36). Shannon diversity ($P = 0.0022$),

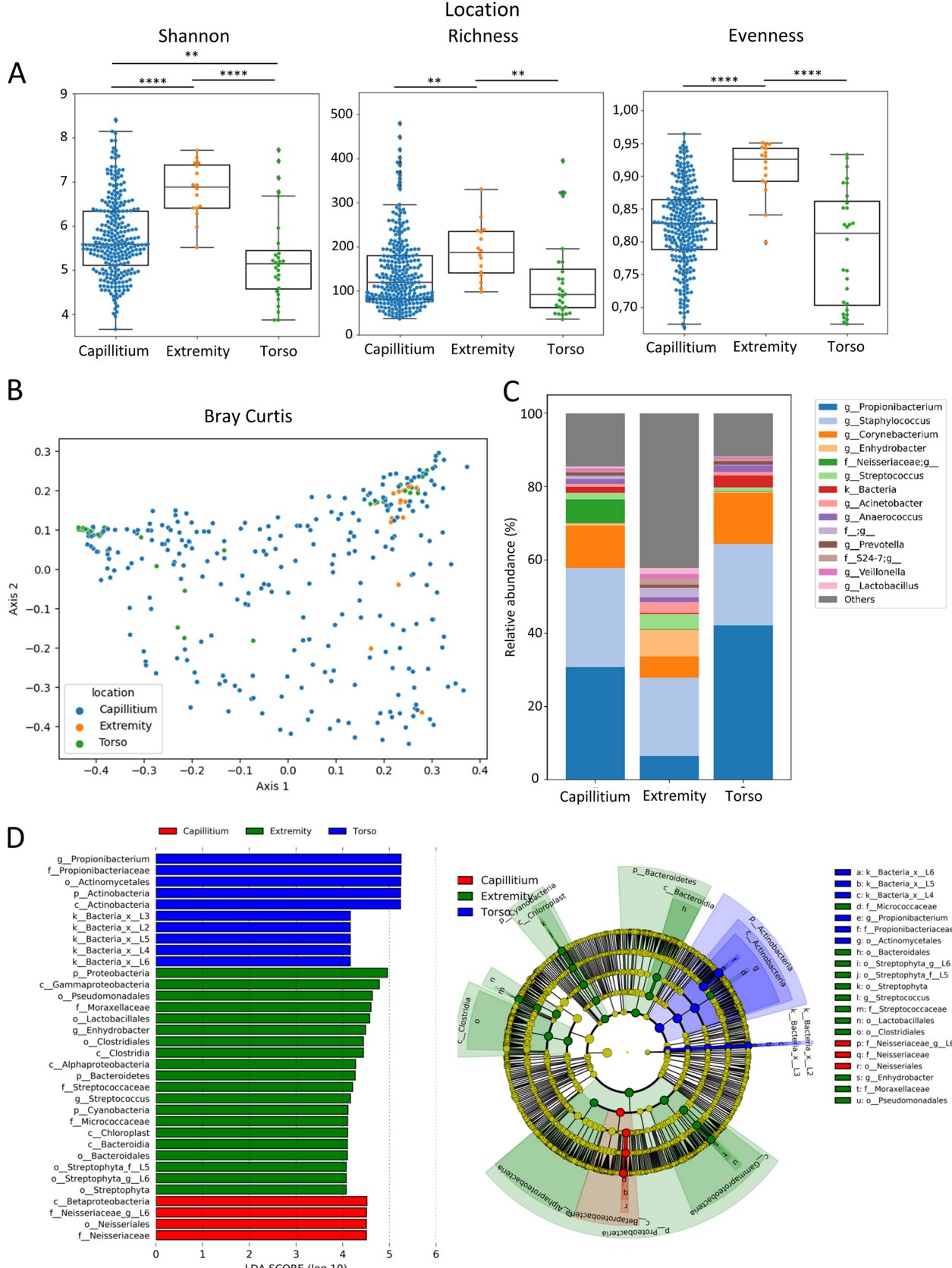

**FIG 1** (A) Shannon diversity, bacterial richness, determined by observed features (amplicon sequence variants [ASV]), and evenness in skin microbiome samples from capillitium (blue), extremity (orange) and torso (green). Kruskal-Wallis test was used to test for significant differences

richness (*P* = 0.0213) and evenness (*P* = 0.0095) were higher at treatment completion (week 24) compared to week 0 before start of therapy (Fig. 2A). Shannon diversity (*P* = 0.0457) and richness (*P* = 0.0465) were significantly lower at week 36 compared with week 24 and were comparable with week 0. No significant differences in Shannon diversity, richness, and evenness were existent between week 0 and 36. Interestingly, the relative abundance of the genus *Staphylococcus* of the skin microbiome of AK lesions before start of therapy (week 0) was significantly higher, with a mean relative abundance of 35.2%, compared to a mean relative abundance of 19.8% at week 24 (*P* = 0.0001) and a mean relative abundance of 22.8% at week 36 (*P* = 0.0001) (Fig. 2B). *Staphylococcus* was the only genus that was identified as biomarker with a linear discriminant analysis (LDA) score >4 and was enriched at week 0 (Fig. 2C). The reduction of the relative *Staphylococcus* abundance at week 24 and 36 was mainly attributed to a reduction of CoNS, while the mean relative *S. aureus* abundance was lower but not significantly different between the different time points (Fig. 2D).

**The relative abundance of CoNS is reduced after CAP and DIC-treatment at week 24 and remains significantly reduced in CAP-treated patients at week 36.** To study whether CAP and DIC treatment may differently affect the skin microbiome of AK patients, we compared the skin microbiome of both treatments for all samples at all time points and in addition also at individual time points. The alpha diversity metrics Shannon index, richness, and evenness did not differ significantly (Fig. 3A), and LEfSe did not identify biomarkers with an LDA score ≥ 3 associated with the treatment at time point of treatment completion (week 24) or 12 weeks after treatment completion (week 36), indicating that the differences are minor (Fig. S1). Notably, the reduction of relative *Staphylococcus* genus abundance was observed for both CAP and DIC treatment and for capillitium samples only (Fig. 3 B and C), which represented the majority of the samples of the study. The reduction of the relative abundance of *Staphylococcus* of *16S rRNA* sequences was due largely to CoNS-reduction and was observed in both CAP- and DIC-treated AK regions (Fig. 3D).

**The skin microbiome of AK at the capillitium responding to CAP at the end of treatment (week 24) is associated with an increased relative abundance of *Corynebacterium*.** Response to therapy at the capillitium (largest subcohort of ACTICAP) was determined directly after end of treatment (week 24), and additionally at the end of the study at week 36 to assess long-term response. Response was assessed for both treatments in each patient separately. A response to treatment (responder, R) was defined by the reduction of the AK area by at least 25% compared to baseline. At time point of treatment completion (week 24), a total of 38 patients were classified as R to CAP and 35 patients were classified as R to DIC treatment. The skin microbiome of AK patients responding to therapy at week 24 was characterized by a significantly higher mean relative abundance of *Corynebacterium* genus of 14.9% at this time point (*P* = 0.0038) compared to those patients who did not respond to CAP treatment (mean 3.98%) (Fig. 4A and B). *Corynebacterium* was the only genus that was identified as significantly different by ANCOM (9) to discriminate R from nonresponder (nR) at week 24 and the only genus identified by LEfSe as biomarker with an LDA score >4 for treatment response at week 24. Interestingly, the relative *Corynebacterium* abundance was increased in patients classified as responders to CAP treatment at week 24 already before the start of the therapy (week 0). They exhibited a mean relative *Corynebacterium* abundance of 12.1% compared to 5.3% in nR at that time point (*P* = 0.0151). In addition, the relative *Corynebacterium* abundance of patients classified as responders at week 24 was also higher 12 weeks after completion of therapy (week 36) with a relative abundance of

**FIG 1** Legend (Continued)

among groups. *, *P* < 0.05; **, *P* < 0.01; ***, *P* < 0.001; ****, *P* < 0.0001. (B) Principal coordinates analysis (PCoA) of skin microbiome samples from capillitium (blue), extremity (orange) and torso (green) of Bray-Curtis distance matrix. PERMANOVA multivariate analysis was used to test for significant differences. (C) Mean relative abundance of the 15 most abundant genera in skin microbiome samples from capillitium, extremity, and torso. (D) Linear discriminant effect size (LEfSe) analysis of bacterial taxa enriched in Capillitium (red), extremity (green) and torso (blue). The cladogram reports the taxa showing different abundance values (LDA score >4) according to LEfSe. AK locations are presented in the color of the most abundant group.

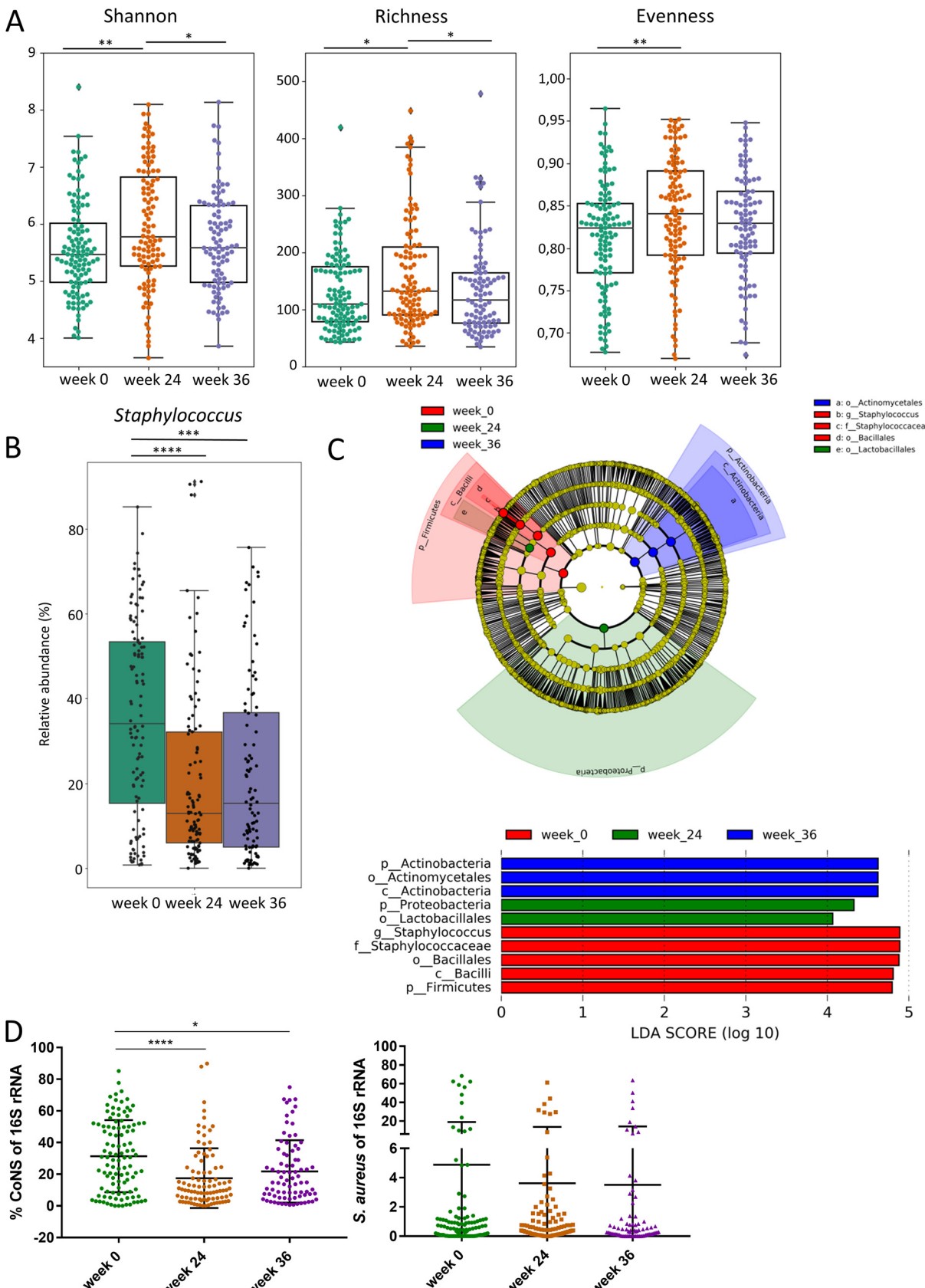

**FIG 2** (A) Shannon diversity, bacterial richness determined by observed features (amplicon sequence variants [ASV]) and evenness in skin microbiome samples at week 0 (green), week 24 (brown) and week 36 (purple). Kruskal-Wallis test was used to test for significant differences

16.80% in R and 4.5% in nR ($P = 0.0078$). *Corynebacterium kroppenstedtii, Corynebacterium bovis,* and two species of the *Corynebacterium* genus that were not identified by the Greengenes database were the four most abundant species accounting for about 98.9% of the capillitium reads that were assigned to bacteria of the *Corynebacterium* genus. Furthermore, *C. durum, C. lubricantis, C. mastitidis, C. simulans, C. stationis,* and *C. variabile* were identified by the Greengenes database from *16S rRNA* gene sequences of the samples of our study.

**The skin microbiome of AK at the capillitium of CAP nonresponders at follow up (week 36) is characterized by a higher relative *S. aureus* abundance compared to responders.** In addition, we compared the skin microbiome of patients that were classified as R 12 weeks after completion of therapy (week 36) with those that were classified as nR at this time point. nR were characterized by a higher relative abundance of *Staphylococcus* genus at week 0 before start of therapy than R (Fig. 5A and B). However, the relative *Staphylococcus* abundance was reduced at week 24 in both, R and nR, and did not significantly differ between both groups at that time point. Interestingly, nR exhibited a significantly higher relative *S. aureus* abundance compared to R at week 36 for both treatments, while the *S. aureus* relative abundance in these patients did not differ significantly at week 0 and 24 (Fig. 5C) The relative abundance of CoNS was significantly higher in CAP-treated nR at week 0 compared to R at that time point but was not significantly different at the other time points between both groups (Fig. 5D).

**The lesional AK microbiome is characterized by a reduction of the total bacterial and *Staphylococcus* loads after treatment and an increased absolute *S. aureus* abundance is a biomarker for CAP long-term nonresponders 12 weeks after the end of treatment.** To determine effects of CAP and DIC treatments on the absolute bacterial abundance, we performed a qPCR assay allowing for the quantification of the *16S rRNA* gene copies. The bacterial load was lower at week 24 ($P = 0.0286$) with a median/mean absolute abundance of 793,252/11,246,154 copies compared with week 0 with a median/mean abundance of 2,186,940/4,594566 (Fig. 6A). Also, the absolute *Staphylococcus* abundance decreased from a median/mean of 586,168/5,303,343 *16S rRNA* copies at week 0 to a median/mean abundance of 99,358/1,774,733 at week 24 ($P = 0.0004$) and the median/mean abundance was 100,290/4,006,945 copies at week 36, which was also significantly lower than the abundance at week 0 ($P = 0.0147$) (Fig. 6B). The decrease of *Staphylococcus* abundance at week 24 was mainly due to a decrease in the absolute CoNS abundance from 471,979 at week 0 to 55,135 at week 24 ($P = 0.0003$) (Fig. 6C), while the absolute *S. aureus* abundance was not significantly different between week 0 and week 24 (Fig. 6D). Interestingly, the absolute *S. aureus* abundance decreased after treatment completion and was lowest at week 36 with a median of 872.3 *16S rRNA* gene copies. The *16S rRNA* gene copy number was significantly lower at week 36 (872.3) compared with week 0 ($P = 0.0217$) with a median *16S rRNA* gene copy number of 6312, suggesting ongoing skin microbiota modulation beyond the treatment period. The median CoNS abundance however increased again between week 24 and week 36. The absolute *Corynebacterium* abundance did not change significantly between the three time points (Fig. 6E), suggesting unequal treatment effects on the absolute abundance for individual species. Although both treatments differ in their mechanism of action, the trends on the reduction of the absolute abundance were similar for both treatments. However, the absolute *Staphylococcus* and CoNS abundances were significantly reduced at week 24 after DIC treatment compared with week 0, while there was a tendency for CAP, which did not reach significance (Fig. S2). In addition, the reduction of the absolute

**FIG 2** Legend (Continued)

among groups. *, $P < 0.05$; **, $P < 0.01$. (B) Relative abundance of *Staphylococcus* genus in skin microbiome samples from week 0, week 24 and week 36. ***, $P < 0.001$; ****, $P < 0.0001$. (C) Linear discriminant effect size (LEfSe) analysis of bacterial taxa enriched in week 0 (red), week 24 (green), and week 36 (blue). The cladogram reports the taxa showing different abundance values (LDA score >4) according to LEfSe. Colors are presented in the color of the most abundant group. (D) Relative abundance of coagulase negative staphylococci (CoNS) and *S. aureus* of *16S rRNA* sequences as determined by quantitative real-time PCR at week 0 (green), week 24 (brown) and week 36 (purple). *, $P < 0.05$; ****, $P < 0.0001$.

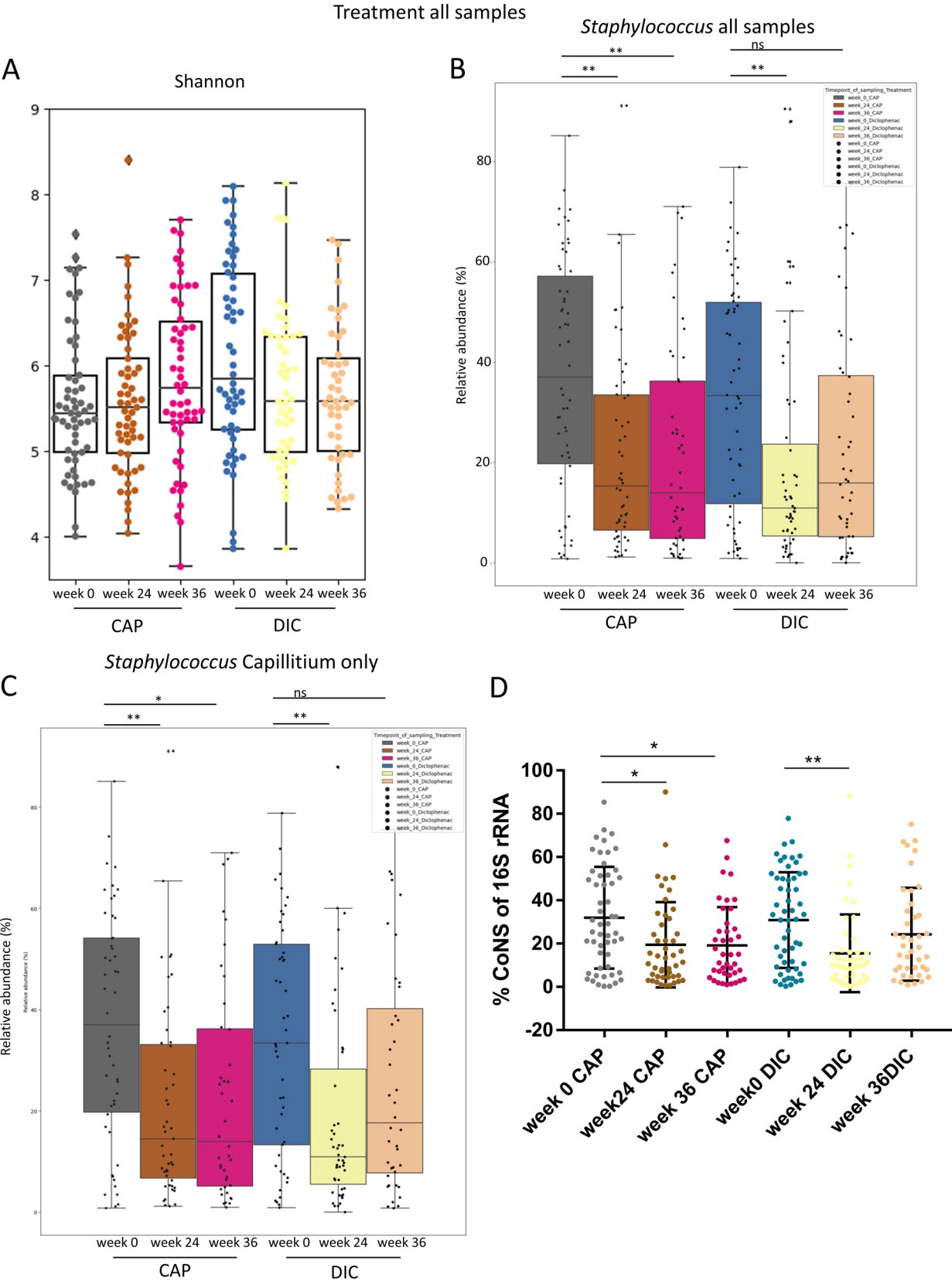

**FIG 3** (A) Shannon diversity in skin microbiome samples receiving CAP versus DIC treatment at week 0, week 24, and week 36. Relative abundance of *Staphylococcus* in all samples (B) and Capillitium samples only (C) receiving CAP versus DIC treatment at week 0, week 24, and week 36. (D) Relative abundance of coagulase-negative staphylococci (CoNS) of *16S rRNA* sequences for week 0, week 24 and week 36 for CAP versus DIC treatment. Kruskal-Wallis test was used to test for significant differences among groups. *, *P* < 0.05; **, *P* < 0.01.

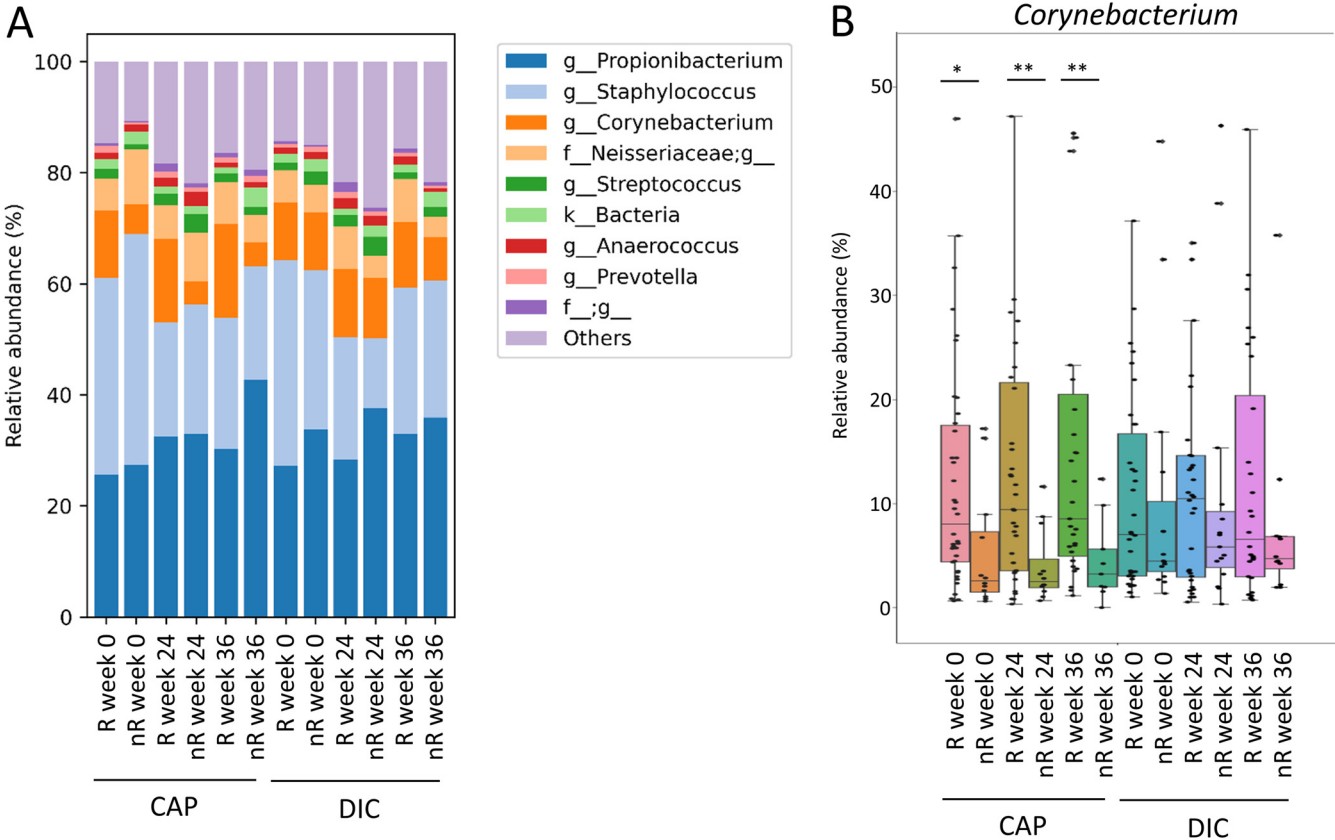

**FIG 4** (A) Mean relative abundance of the 10 most abundant genera in skin microbiome samples from capillitium of individuals classified as responders (R) and nonresponders (nR) for CAP or DIC treatment at week 24 for all time points (week 0, week 24, and week 36). (B) Mean relative *Corynebacterium* genus abundance of patients classified as R and nR for CAP or DIC treatment individually at week 24. Relative *Corynebacterium* abundance is shown for all time points of R and nR.

*S. aureus* abundance was significant at week 36 for DIC only (Fig. S2). A tendency for higher absolute abundances of the *Corynebacterium* genus was present for patients classified as R for week 24 (Fig. S3) and week 36 (Fig. 7) compared with nR for the majority of time points. However, a significant higher *Corynebacterium* abundance was observed only for DIC treated patients classified as R at week 36 compared with nR for the time point week 24 (Fig. 7). Interestingly, the absolute *S. aureus* abundance was significantly higher for CAP treated patients classified as nR at follow-up (week 36) compared to R at that time point (Fig. 7), indicating that proliferation of *S. aureus* 12 weeks after CAP therapy completion is a biomarker for long-term treatment failure.

## DISCUSSION

In our longitudinal prospective study, the skin microbiome of AK patients differed significantly between AK lesions of the capillitium, torso, and extremities. The highest diversity was present in AK lesions of the extremities. We showed that the relative abundance of the *Staphylococcus* genus was significantly higher before start of therapy compared with the time points of end of therapy at week 24 and at week 36 for both, CAP and DIC therapy. The reduction of *Staphylococcus* relative abundance was due to a reduction of the relative CoNS abundance and observed in both, CAP and DIC treated patients. The skin microbiome of patients classified as R at time point of CAP therapy completion was characterized by a higher relative abundance of *Corynebacterium* genus at all investigated time points, also before start of therapy. The skin microbiome of patients classified as nR at week 36 exhibited a higher relative abundance of *S. aureus* than R at week 36. In addition, the total bacterial and *Staphylococcus* loads were higher before therapy initiation than at time point of therapy completion; and absolute *S. aureus* abundance

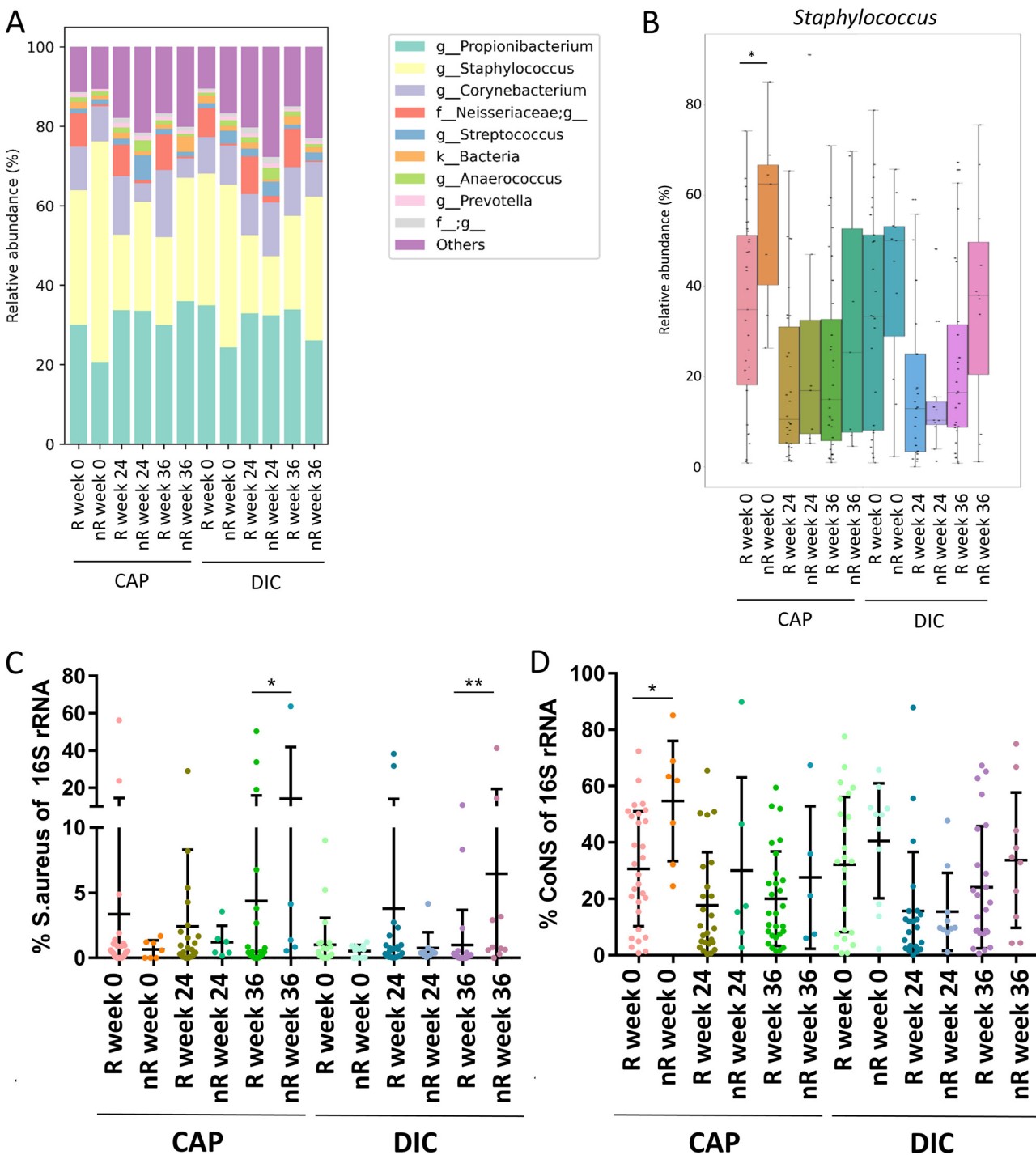

**FIG 5** (A) Mean relative abundance of the 10 most abundant genera in skin microbiome samples from capillitium of individuals classified as responders (R) and nonresponders (nR) for CAP or DIC treatment at week 36 for all time points (week 0, week 24, and week 36). (B) Mean relative *Staphylococcus* genus abundance of patients with capillitium AK classified as responders (R) and nonresponders (nR) for CAP or DIC treatment individually at week 36. Relative *Staphylococcus* abundance is shown for all time points of R and nR. (C) Relative abundance of *Staphylococcus aureus* (*S. aureus*) in capillitium samples of patients classified as R and nR for CAP or DIC treatment at week 36. The relative *S. aureus* abundance in these patients is shown for all time points. (D) Relative abundance of coagulase-negative staphylococci (CoNS) in capillitium samples of patients classified as R and nR for CAP or DIC treatment at week 36. The relative CoNS bundance in these patients is shown for all time points.

12 weeks after therapy completion was higher in CAP treated patients classified as nR than in R at that time point.

 The differences associated with the location of the AK lesions mimic those described for these skin microbiome regions in healthy individuals, indicating that the diseased

## Timepoint Capillitium samples

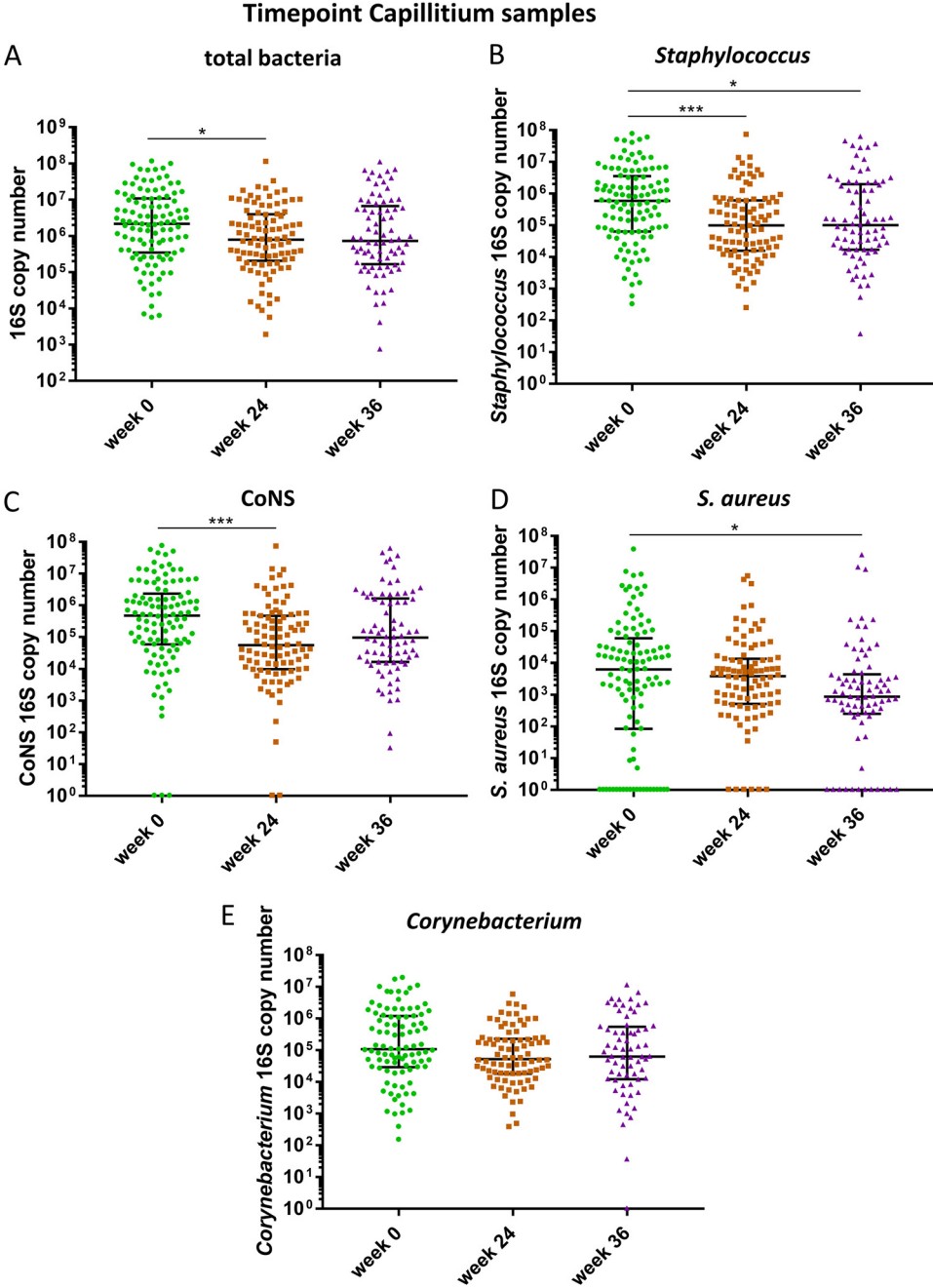

**FIG 6** Absolute abundance of bacteria in individual samples assessed by qPCR quantifying the *16S rRNA* gene copy number in skin swab samples from capillitium at week 0 (green), week 24 (brown) and week 36 (purple). (A) total bacterial load, (B) *Staphylococcus* genus load, (C) coagulase negative staphylococci, (CoNS) (D) *S. aureus*, (E) *Corynebacterium* genus. Bars indicate median ± interquartile range. Kruskal-Wallis test was used to test for significant differences among groups. *, $P < 0.05$; **, $P < 0.01$; ***, $P < 0.001$.

skin microbiome is not basically altered and the regional microbiota characteristics are maintained in AK areas. The back of the torso is a sebaceous area characterized by a low diversity with *Cutibacterium* (*Propionibacterium)* as the dominant genus in healthy individuals (4). In AK patients of our study, the torso was the site with the lowest alpha diversity metrics and the highest relative abundance of *Cutibacterium*—the dominating genus of the skin microbiome of this site. The skin microbiome of AK lesions from extremities in our study showed the highest diversity and richness of the three body regions and was enriched in Proteobacteria, *Bacteroidia,* and *Clostridiales* among other taxa. This is in

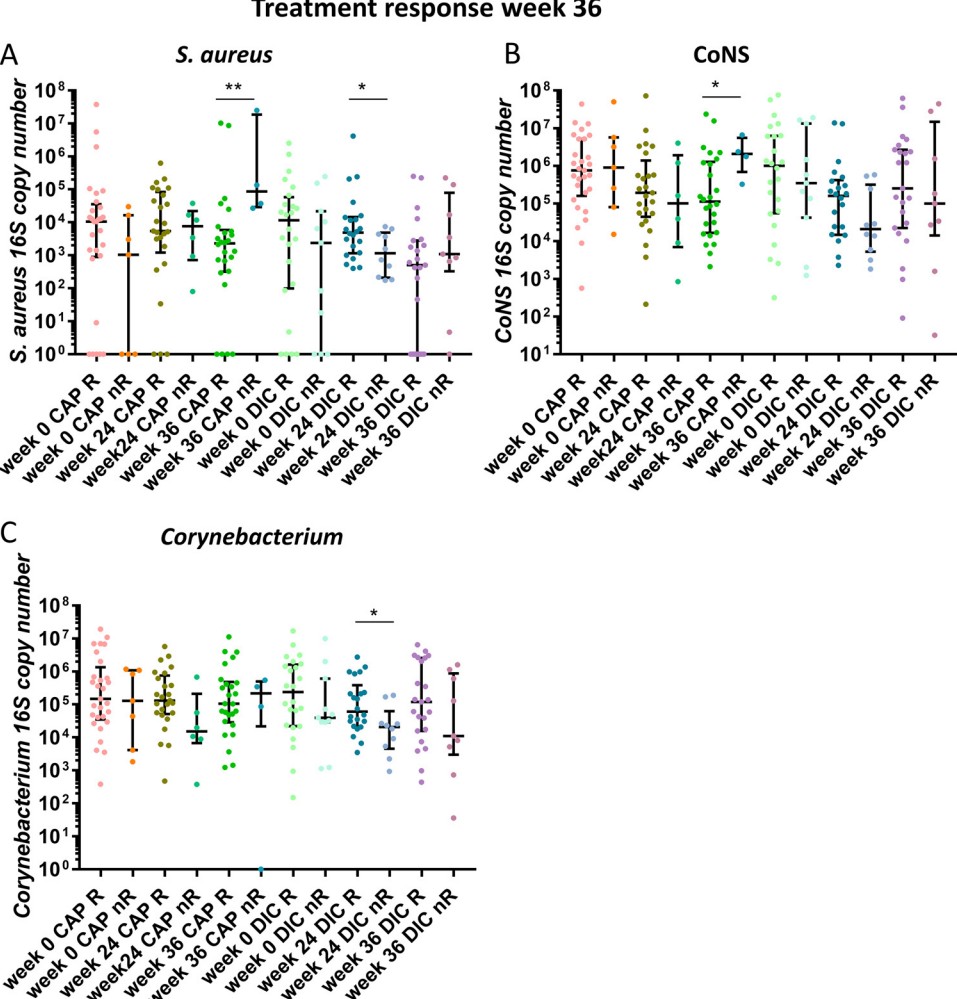

**FIG 7** Absolute *16S rRNA* gene copy number from capillitium skin swab samples linked to treatment response at week 36. The relative abundance of the respective bacteria for patients classified as R versus nR at week 36 is shown for all time points (week 0, 24 and 36) (A) *S. aureus*, (B) coagulase negative staphylococci (CoNS), (C) *Corynebacterium* genus. Bars indicate median $\pm$ interquartile range. Mann-Whitney U test was performed to test for significant differences between R and nR for the respective time point and treatment.

line with previous studies that observed that dry skin areas of healthy individuals, that include large parts of the extremities, are known to harbor the greatest diversity of the skin microbiome (4, 10). This location is also colonized by Gram-negative bacteria, including Proteobacteria, and has a phylogenetic diversity that may even exceed the diversity of the oral cavity and gut of these individuals (4).

In our study, the genus *Staphylococcus* was the second most abundant genus of the skin microbiome of AK patients after *Cutibacterium* (*Propionibacterium)* and the genus with the most pronounced alterations of the relative abundance over the different time points of treatment. The relative abundance of the genus *Staphylococcus* has repeatedly been found to be increased in AK and SCC patients (6, 11, 12). It has been suggested that *Staphylococcus*, particularly *S. aureus*, is involved in AK pathogenesis and may play a protumorigenic role, e.g., by stimulating the production of beta-defensins of keratinocytes that have antibacterial properties, but may increase proliferation of keratinocytes in an autocrine manner (6). Besides, *S. aureus* can promote skin inflammation by producing toxins that may also mediate protumorigenic inflammatory responses (13). The relative *Staphylococcus* abundance was significantly reduced at the end of both treatments, CAP and DIC, and was significantly lower in R compared to nR 12 weeks after treatment completion, indicating that both treatments affect the skin

microbiome composition similarly. The reduction of the relative *Staphylococcus* abundance was largely due to the reduction of the relative abundance of CoNS. Although *S. aureus* is the pathogen most commonly reported to promote tumor progression in skin malignancies, CoNS has been found to be overrepresented in transplant patients with SCC and was linked to tumor progression (14). However, the long-term response toward treatment at week 36 in our study was associated with a lower relative and absolute *S. aureus* abundance in R than in nR, supporting a relevance of *S. aureus* for this disease and suggesting that an increase of *S. aureus* after completion of treatment is a biomarker for long-term treatment failure. The formation of microbial communities, their function and stability depend on host factors and the interaction between these microbes and effects may be context specific. Interactions between microbes can be competitively or synergistically. While the growth of *S. aureus* is prohibited, e.g., by the production of lugdunin by *Staphylococcus lugdunensis* (15), some *Cutibacterium* species induce the aggregation of *S. aureus* and its biofilm formation (16). In our study, the relative abundance of *Corynebacterium* genus was higher in R than in nR at the end of CAP treatment, and the skin microbiome of R was also characterized by a higher relative *Corynebacterium* abundance compared to that of nR before start of CAP treatment. *Corynebacterium* genus has been shown to shift *S. aureus* toward commensalism by reducing its virulence factor expression (17), which might be exploited to modulate *S. aureus* behavior in future therapies. Shifting *S. aureus* toward commensalism was observed for several different commensal *Corynebacterium* species by others (17), indicating a genus specific rather than species specific potential. In our study, sequencing reads were assigned to several species; however, four *Corynebacterium* species, two that could not be identified to the species level using *16S rRNA* gene sequencing and the two species *C. kroppenstedtii* and *C. bovis*, were responsible for more than 98% of the sequences of the *Corynebacterium* genus.

The total bacterial loads were significantly lower at the end of treatment (week 24) compared to week 0, suggesting that antibacterial effects of CAP and DIC treatments reduce bacterial load of AK skin lesions. The total abundance was not reduced equally for all bacterial species, indicating differential susceptibility toward treatments. Several mechanisms have been reported for antibacterial CAP activity. Reactive oxygen and nitrogen species produced by CAP can trigger programmed cell death with hallmarks of apoptosis by short-term treatment, but long-term CAP treatment can physically destruct intracellular components (18). In addition, CAP may promote the potency of macrophages to kill *S. aureus* by improving the formation of degradative phagosomes (19). Besides the anti-inflammatory effects by inhibiting the prostaglandin synthesis, DIC was suggested to affect *in vitro* bacterial growth by modulating DNA synthesis (20). Despite the different nature of both treatments, the differences on the skin microbiome composition linked to the treatments in our study were minor. However, absolute abundance of *Staphylococcus* and CoNS were significantly lower at week 24 than at week 0 for DIC but not CAP. Also, the reduction of absolute *S. aureus* abundance at week 36 was significant for DIC but not CAP. Patients classified as nR 12 weeks after end of CAP therapy were characterized by significantly higher *S. aureus* relative and absolute abundance than R, indicating that a high *S. aureus* abundance may be a biomarker for CAP treatment failure. Although most studies have focused on *S. aureus* as most important species of the *Staphylococcus* genus, a recent report links *S. epidermidis*, a common species of CoNS, to SCC progression in transplant patients (14). Pathologies might be context dependent. Pathogens like *S. aureus* may asymptomatically colonize the skin, whereas mutualists like *S. epidermidis* or corynebacteria sometimes then promote disease (17, 21, 22).

This study is limited by its monocentric design. Although the longitudinal design of our study investigating the skin microbiome in patients over a period of about 9 months, we cannot exclude that other unconsidered factors contribute to the differences in the skin microbiome observed in our study.

**Conclusions.** Our study reveals a reduction of the absolute bacterial abundance of AK lesions and alterations of the AK lesional microbiome composition after field-directed therapies. The microbiome alterations affect bacteria that are considered to be involved in pathogenesis of skin malignancies. The reduction of the overabundance of *Staphylococcus* of the skin microbiome after treatment of AK lesions, the persistent lower relative *Staphylococcus* genus und relative and absolute *S. aureus* abundance in long-term responders, and the link of *Corynebacterium* genus with response to CAP treatment encourage further studies to investigate the role of the skin microbiome for the carcinogenesis of epithelial skin cancer as well as its function as predictive therapeutic biomarker in AK.

## MATERIALS AND METHODS

**Study population.** We included 59 AK patients of the ACTICAP study, a prospective randomized study that was performed between 2017 and 2019 (8). The Ethics Committee of the University Hospital Essen reviewed and approved the study (17-7546-BO). The study was performed in accordance with the latest version of the Declaration of Helsinki. All patients signed written informed consent.

**Sampling and sample processing.** Two anatomically comparable areas of approximately 19.6 cm$^2$ exhibiting similar numbers of AK at baseline (at least $n = 2$) were chosen for sampling in each participant. The ACTICAP study design included a washout phase of potentially prohibited medication, including photosensitizing agents, immunomodulators, antineoplastic agents and systemic retinoids for 4 to 12 weeks before treatment with CAP or DIC was started, depending on drug type. Patients were included after a minimum of 6 months after physical procedures (e.g., surgery, laser, PDT). The identified treatment areas were computationally randomized to either CAP that was applied for 180 s twice a week or topical DIC (diclofenac 3%) twice a day (8). Each area was sampled at three time points: first, before start of treatment (week 0); second, at the end of the treatment period (week 24); third, 12 weeks after completion of the treatment period at follow-up (week 36). Of 59 patients, 51 patients had AK of the capillitium, 5 of the torso and 3 of extremities (two patients with AK of the hands, one patients with AK of the lower leg). The clinical diagnosis had been confirmed by routine histology previously. Skin swabs were taken using Puritan sterile standard foam swabs (Puritan Medical Products, Guilford, ME, US) moistened with sterile SCF-1 solution (5 mm Tris buffer, 1 mM EDTA and 0.5% Tween 20) by rubbing the swabs about 50 times back and forth at the site of AK for 30 s. The swabs were placed in PowerBead Tubes of the DNeasy PowerLyzer PowerSoil kit (Qiagen, Hilden, Germany) filled with 250 $\mu$L SCF1 solution. Samples were frozen at $-20$°C until further processing. In addition, swabs not used for sampling were shaken in SCF1 solution as negative-control samples and were prepared in parallel with the patient samples. Sixteen patients were lost for the follow-up examination, unrelated to treatment but mainly due to incompliance. Extraction of microbial DNA was performed using the DNeasy PowerLyzer PowerSoil kit (Qiagen) according to the manufacturer¨s guidelines. DNA was eluted in 30 $\mu$L TE buffer. Amplicon PCR was performed as described previously (23) amplifying the V3/V4 *16S rRNA* gene region with 25 cycles. We used the Nextera XT Index kit V2 primers for index PCR. Sequencing was performed using the Illumina MiSeq with the 600 cycle reagent kit. Swabs shaked in SCF1 solution and a PCR water sample were used as negative controls.

For absolute quantification of microbes in individual samples, we used the Femto Bacterial DNA Quantification kit (Zymo Research, Freiburg, Germany) according to the manufacturer¨s guidelines. This qPCR assay includes bacterial standards, allowing for determination of *16S rRNA* gene copy numbers in individual samples. As the DNA of all swab samples had been eluted in a volume of 30 $\mu$L and 2 $\mu$L were used for individual PCRs, the resulting *16S rRNA* gene copies of individual samples calculated from the standard curves were multiplied with 15 to obtain the copy number/sample.

**Microbiome data analysis.** The QIIME2 (Quantitative Insights Into Microbial Ecology2) pipeline (24) was used for analysis of fastq files with the DADA2-package integrated in QIIME2 to correct sequences and to filter chimeric sequences using the consensus-method. A total of 4,076,151 quality filtered reads representing 20,314 amplicon sequence variants (ASV) were obtained from 321 patient samples and eight negative-control samples. Of 321 patient samples, a total of 4,053,981 quality filtered reads represented 20,222 ASV. For the identification of contaminating ASVs, we used the Decontam software package in R (25) using the "prevalence" method, that identified 124 contaminating ASVs. Contaminating ASVs were removed from all samples, resulting in a total of 3,969,719 quality filtered reads from 321 samples, including 20,098 ASVs. Four patient samples with a frequency $\leq$ 2,000 quality filtered reads were removed from data analysis. The remaining 317 samples represented 20,083 ASVs and 3,967,062 quality filtered sequences with a mean and median frequency of 12,514 and 11,944 reads and a minimum of 2,065 reads.

The q2 diversity plugin was used for computing alpha and beta diversity metrics. We used the q2-feature-classifier plugin for taxonomy assignment with a Naïve Bayes classifier, trained on the Greengenes 13_8_99% OTUs *16S rRNA* gene full-length sequences. Comparisons between categorical metadata columns and alpha diversity metrics were computed using the *qiime diversity alpha-group-significance* command with Kruskal-Wallis test. To test for significant differences in beta diversity among groups, we used the *qiime diversity beta-group significance* command with permutational multivariate analysis of variance (PERMANOVA) of distance matrices with 999 permutations. ANCOM (9) and Kruskal–Wallis test was used for analysis of differences in the relative taxa abundance. Biomarkers were assessed using LEfSe (26) with a $P < 0.05$ for bacterial class comparison using Kruskal-Wallis test and a linear discriminant analysis (LDA) score $>$4.

Patient characteristics were tested using $t$ test for metric variables and Fisher exact test for categorical variables with a significance level <0.05. Alpha diversity metrics, barplots, boxplots and principle coordinates analysis (PCoA) were visualized using Dokdo, Version 1.7 (https://github.com/sbslee/dokdo). Dotplots for assessing absolute bacterial abundances and *S. aureus* and CoNS relative abundance were visualized using GraphPad Prism Ver. 7 software (San Diego, CA, US).

**TaqMan PCR for quantification of the relative *S. aureus* and CoNS abundance.** *S. aureus* and other members of the *Staphylococcus* genus are inadequately classified to the species level using 16S rRNA gene sequencing (27). The TaqMan assay described here allows for the quantification of the relative abundance of *S. aureus* from staphylococci present in microbial communities using two MGB-probes in a single PCR tube for relative quantification, as described elsewhere (28, 29). We used TaqMan probes, that had been validated previously to specifically amplify and bind the *tuf* gene (30) of either *S. aureus* (FAM labeled) or *Staphylococcus* genus (HEX labeled). The relative abundance of *S. aureus* to *Staphylococcus* genus was determined using a calibration curve of several mixtures of *S. aureus* (DSMZ 799) and *S. epidermidis* (DSMZ 1498). For the generation of the mixtures, *S. aureus* and *S. epidermidis*, single colonies grown on Columbia blood agar were subcultured for 6 h in brain heart infusion (Oxoid, Wesel, Germany). Dilutions in PBS from $10^{-1}$ to $10^{-7}$ were plated on Columbia blood agar and incubated at 37°C with 5% $CO_2$ atmosphere. After 24 h, CFU (CFU) were counted to determine the concentration of *S. aureus* (Deutsche Sammlung von Mikroorganismen und Zellkulturen [DSMZ] 799) and *S. epidermidis* (DSMZ 1498) in the pure cultures to generate various mixtures. Besides the single strain cultures, various *S. aureus*/*S. epidermidis* mixtures of the reference isolates were generated for the calibration curve with a relative *S. aureus* abundance ranging between 1% and 99.9%. DNA of the calibration standards was extracted using the ZymoBIOMICS DNA Miniprep kit (Zymo Research, Freiburg, Germany) according to the manufacturer's recommendations. Mixtures were subjected to NGS deep amplicon sequencing to determine the precise composition of the relative *S. aureus* abundance of the mixtures. The TaqMan PCR was run on a Roche LightCycler 480 (Roche, Basel, Swiss) with the LightCycler 480 Probes MasterMix in a reaction volume of 20 $\mu$L in a two-step PCR with 40 cycles at 60°C for 60 s. Calibration standards were used in duplicate and included in each LightCycler run. All samples were analyzed in duplicates. Data analysis was performed using the LightCycler 480 (Roche) software using the "fixed-points" method. Of 33 samples, the relative abundance of *S. aureus* and CoNS could not be calculated due to low total amount of bacterial DNA of the samples. Those were excluded from analyses. The relative CoNS abundance was calculated as follows: relative CoNS abundance = relative *Staphylococcus* abundance – relative *S. aureus* abundance.

**Data availability.** Sequencing data are available in the National Center for Biotechnology Information Sequence Read Archive under BioProject accession ID PRJNA894468.

## SUPPLEMENTAL MATERIAL

Supplemental material is available online only.
**SUPPLEMENTAL FILE 1**, PDF file, 0.8 MB.

## ACKNOWLEDGMENTS

This study was technically advised by Mary McGovern and Jeiram Jeyaratnam and financially supported in parts by Adtec Europe Limited.

We have no conflicts of interest to declare.

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
