## [Reviewer comments · Microbiology Spectrum]

Microbiology Spectrum

Reduced *Staphylococcus* abundance characterizes the lesional microbiome of actinic keratosis patients after field-directed therapies

Jan Kehrmann, Fiona Koch, Skrollan Zumdick, Anna Höwner, Lara Best, Lars Masshoefer, Sarah Scharfenberg, Michael Zeschngk, Jürgen Becker, Dirk Schadendorf, Jan Buer, and Alexander Rösch

Corresponding Author(s): Jan Kehrmann, Universitat Duisburg-Essen

Review Timeline:

Submission Date:	November 4, 2022
Editorial Decision:	January 21, 2023
Revision Received:	March 20, 2023
Accepted:	March 23, 2023

Editor: Victor Gonzalez

Reviewer(s): Disclosure of reviewer identity is with reference to reviewer comments included in decision letter(s). The following individuals involved in review of your submission have agreed to reveal their identity: Gregor Gorkiewicz (Reviewer #1)

Transaction Report:

DOI: <https://doi.org/10.1128/spectrum.04401-22>

January 21, 2023

Dr. Jan Kehrmann
University Hospital Essen, University of Duisburg-Essen
Institute of Medical Microbiology
Virchowstr. 179
Essen 45147
Germany

Re: Spectrum04401-22 (Reduced *Staphylococcus* abundance characterizes the lesional microbiome of actinic keratosis patients after field-directed therapies)

Dear Dr. Jan Kehrmann:

Link Not Available

Sincerely,

Victor Gonzalez

Journals Department
Reviewer comments:

Reviewer #1 (Comments for the Author):

Kehrmann et al. described skin microbiome changes upon topical treatments of AK patients in a longitudinal manner. 321 samples originating from 59 AK patients were analyzed. Treatment were DIC vs CAP on 2 different lesions per patient. The major findings were that both treatments reduced *Staphylococcus* (mainly CoNS) and non-responders showed higher *S. aureus* relative abundance at follow up. In general, this is a well performed interesting study specifying topical treatment effects on the skin microbiome of AK patients. I would suggest to (i) also use qPCR for *Staphylococcus*, *S. aureus* and *Corynebacterium* and show this in terms of loads rather than just relative abundances and (ii) authors should specify which *Corynebacterium* spp. are the ones enriched in CAP treatment

major

line 118 ff "The reduction of the Staphylococcus abundance was mainly attributed to a reduction of CoNS, while the mean relative S. aureus abundance was lower but not significantly different between the different timepoints (Figure 2D)." Change wording to relative abundance (& throughout the manuscript where appropriate); how does this translate into absolute abundance? of Staphylococcus vs. S. aureus? Important given the pro-tumorigenic role of S. aureus

"Section increased Corynebacterium relative abundance at capillitium in responders"; how does this translate to absolute abundance? could the authors give a little bit more information about the species of Corynebacterium? was it just one species or different ones?

The differences between the two treatments (DIC vs CAP) on the skin microbiome could be described in an individual paragraph for clearness

minor
wording

line 117 "Staphylococcus was the only genus taxon" remove taxon since Staphylococcus per se is specified as a genus (use this also throughout the manuscript where appropriate e.g. line 149)

line 129ff LDA score {greater than or equal to} 3 .. not shown.... please show this data in the supplemental material

Propionibacterium I assume Cutibacterium (which is the current correct designation)

Discussion

Authors should elaborate more on the different pathogenic vs physiologic roles of S. aureus vs. CoNS (e.g. S. epidermidis) vs. Corynebacterium and their relation to the described treatment effects; there is a plethora of literature out investigating these taxa in the context of skin diseases, tumors and inflammation. How does this fit to their findings? Is the reduction of certain taxa (or increase) somehow mechanistically beneficial (e.g. by increasing beneficial taxa by the treatments) or is the harsh treatment just killing certain taxa? How acts DIC on skin microbiota directly or via the immune-system?

Reviewer #2 (Comments for the Author):

In this manuscript, the use of diclofenac (DIC) and cold atmospheric plasma (CAP) on the skin microbiome in the treatment of actinic keratosis was studied.

The following comments are made:

1. Line 36. "tuff" is a gene? If so, put it.
2. Lines 86, 245-246. How did the treatments with CAP and DIC? You don't say it. It is a fundamental part of your work, there should be a section to explain the treatments.
3. Line 90. What extremities are you talking about? explain it
4. Line 100. What does "LefSe" mean? put it
5. Line 132-133. "The reduction of the relative abundance of Staphylococcus of 16S rRNA sequences was due largely to CoNS-reduction". It is not clear why you say they are CoNS. How did you measure and verify it? explain it.
6. Line 150. In line 142-143, it is well established that it is an R, but in line 150 nR appears and you do not say what it means or what its definition is. Correct it and define what is an nR.
7. Line 163. "NR", should be nR
8. Line 179-183. What interpretation do you give to these results, discuss it.
9. Line 194. Capitalizing Gram is a proper name.
10. You do not discuss the effect of CAP on the microorganisms found and their differences. Discuss the mechanism of CAP on bacteria and why the differences found.
11. Lines 306-307. What control strains of S. aureus and S. epidermidis did you use?
12. Line 387 and Figure 5. Staphylococcus aureus abbreviated as S. aureus

Staff Comments:

Preparing Revision Guidelines

To submit your modified manuscript, log onto the eJP submission site at <https://spectrum.msubmit.net/cgi-bin/main.plex>. Go to Author Tasks and click the appropriate manuscript title to begin the revision process. The information that you entered when you

first submitted the paper will be displayed. Please update the information as necessary. Here are a few examples of required updates that authors must address:

Please return the manuscript within 60 days; if you cannot complete the modification within this time period, please contact me. If you do not wish to modify the manuscript and prefer to submit it to another journal, please notify me of your decision immediately so that the manuscript may be formally withdrawn from consideration by Microbiology Spectrum.

Point-to-point Reply

We are grateful for the positive and helpful reviews that helped to clearly improve the manuscript. We have complied with all comments and answer them in the following point-to-point reply.

Reviewer #1 (Comments for the Author):

Kehrmann et al. described skin microbiome changes upon topical treatments of AK patients in a longitudinal manner. 321 samples originating from 59 AK patients were analyzed. Treatment were DIC vs CAP on 2 different lesions per patient. The major findings were that both treatments reduced Staphylococcus (mainly CoNS) and non-responders showed higher S. aureus relative abundance at follow up. In general, this is a well performed interesting study specifying topical treatment effects on the skin microbiome of AK patients. I would suggest to (i) also use qPCR for Staphylococcus, S. aureus and Corynebacterium and show this in terms of loads rather than just relative abundances and (ii) authors should specify which Corynebacterium spp. are the ones enriched in CAP treatment

Answer from the authors: Thank you for the summary, your positive feedback and valuable suggestions. We have now included all your suggestions and performed qPCR for determining absolute abundances for individual samples. With the information regarding the total bacterial abundance of each sample and the relative abundances of our 16S rRNA sequencing data, we calculated the absolute abundances of different bacterial genera or species of interest. We have supplemented our revised version with absolute *Staphylococcus*, *S. aureus*, CoNS and *Corynebacterium* abundances as detailed described below and in the revised version of our manuscript. We also describe which *Corynebacterium* spp. were identified from amplicon sequence variants of 16S rRNA gene sequencing. Both is described in your comments below and in the revised version of the manuscript in detail.

major

line 118 ff "The reduction of the Staphylococcus abundance was mainly attributed to a reduction of CoNS, while the mean relative S. aureus abundance was lower but not significantly different between the different timepoints (Figure 2D)." Change wording to relative abundance (& throughout the manuscript where appropriate); how does this translate into absolute abundance? of Staphylococcus vs. S. aureus? Important given the pro-tumorigenic role of S. aureus

Answer from the authors: We have changed the wording into relative abundance throughout the manuscript where appropriate. In addition, we have now performed qPCR to determine absolute abundances using the Femto Bacterial DNA quantification kit (Zymo Research, Freiburg, Germany), which allows for calculation of the 16S rRNA gene copy numbers in individual samples (described in the Materials and Methods section, lines 380-386 of the clean version). The total number of 16S rRNA copies is significantly lower at week 24 compared to week 0, before start of treatment (Figure 6A). Interestingly, the absolute *Staphylococcus* and CoNS abundances are both significantly lower at week 24 compared with week 0. For week 36, only the absolute *Staphylococcus* abundance but not CoNS abundance is significantly lower compared with week 0 (Figure 6B and 6C). However, the *S. aureus* load decreases also after treatment completion and the abundance is lowest at week 36 and significantly lower at that time point compared with week 0 (Figure 6D), indicating that

not only killing effects of the treatments are responsible for *S. aureus* reduction observed, but that also other factors, like (modulation of immune and other host cells) might play a role in microbiome alterations, as the absolute *S. aureus* abundance is even lower 12 weeks after completion of treatments compared to the end of treatment period. *S. aureus* has been considered as pathogen promoting tumor growth, which is in contrast to the CoNS, which had for long periods been regarded as non- or less pathogenic. However, although considered as less pathogenic, CoNS was linked to disease progression in transplant patients with SCC, that exhibit an overabundance of *S. epidermidis* (Krueger A et al. 2022. Changes in the skin microbiome associated with squamous cell carcinoma in transplant recipients. ISME Communications 2:13.). We have included these issues in the results and discussions section, lines 204-223 and in the discussion of the revised manuscript.

"Section increased Corynebacterium relative abundance at capillitium in responders"; how does this translate to absolute abundance? could the authors give a little bit more information about the species of Corynebacterium? was it just one species or different ones?"

Answer from the authors:

We thank the reviewer for this comment. In contrast to the *Staphylococcus* absolute abundance, that significantly decreased at treatment completion compared to week 0, the absolute *Corynebacterium* abundance was not significantly lower at therapy completion (week 24) and 12 weeks after therapy completion (week 36) compared to week 0 before start of treatment, indicating that treatment effects do not affect all bacteria equally (Figure 6E). A tendency for higher absolute abundances of the *Corynebacterium* genus was present for patients classified as R for week 24 (Supplemental Figure 3) and week 36 (Figure 7) compared with nR for the majority of time points. However, a significant higher *Corynebacterium* abundance was observed only for DIC treated patients classified as R at week 36 compared with nR for the timepoint week 24 (Figure 7). We have included these results in the manuscript (lines 223-227 and lines 231-236)

Identification of bacteria to the species level using *16S rRNA* gene sequencing is not always possible and the databases may not always reliably identify the bacteria to the species level. However, among the bacteria of the *Corynebacterium* genus, *Corynebacterium kroppenstedtii*, *Corynebacterium bovis* and two species of the *Corynebacterium* genus that were not identified by the Greengenes database were the four most abundant species accounting for about 98.9% of the capillitium reads that were assigned to bacteria of the *Corynebacterium* genus. Furthermore, reads of the *Corynebacterium* species *C. durum*, *C. lubricantis*, *C. mastitidis*, *C. simulans*, *C. stationis* and *C. variabile* were identified using the Greengenes database. We have now included these information in the results section, lines 184-189 and discussed the results line 298-305.

The differences between the two treatments (DIC vs CAP) on the skin microbiome could be described in an individual paragraph for clearness:

Answer from the authors: Thank you for this suggestion. We have performed analyses regarding the differences in the skin microbiome between CAP vs DIC treatment for the time points at week 24 and 36. Interestingly the differences in the skin microbiome composition between both treatments were minor and LEfSe did not identify biomarkers linked to the treatment with LDA score >3. We have now supplemented the manuscript with the additional absolute abundance data and as the differences between both treatments were minor, we prefer not to add another paragraph but we have described the differences between both treatments in the existing paragraphs in the results section (lines 149-162 and lines 226-239).

minor

wording

line 117 "Staphylococcus was the only genus taxon" remove taxon since Staphylococcus per se is specified as a genus (use this also throughout the manuscript where appropriate e.g. line 149)

Answer from the authors: We have corrected this issue as suggested throughout the manuscript now.

line 129ff LDA score {greater than or equal to} 3 .. not shown.... please show this data in the supplemental material

Answer from the authors: We thank the reviewer for this comment. LEfSe did not identify a biomarker with LDA score ≥ 3 . But we have now added Supplemental Figure 1 showing taxa with an LDA score ≥ 2 and <3 between CAP and DIC treatment for week 24 and 36.

Propionibacterium I assume Cutibacterium (which is the current correct designation)

Answer from the authors: We agree with the reviewer that *Cutibacterium* is the correct nomenclature of the genus previously designated as *Propionibacterium*. However, the Greengenes database used *Propionibacterium* for the taxonomy assignment. We have now replaced *Propionibacterium* by *Cutibacterium* in the manuscript text and explained the former nomenclature in brackets, see line 123 of the revised version of the manuscript.

Discussion

Authors should elaborate more on the different pathogenic vs physiologic roles of S. aureus vs. CoNS (e.g. S. epidermidis) vs. Corynebacterium and their relation to the described treatment effects; there is a plethora of literature out investigating these taxa in the context of skin diseases, tumors and inflammation. How does this fit to their findings? Is the reduction of certain taxa (or increase) somehow mechanistically beneficial (e.g. by increasing beneficial taxa by the treatments) or is the harsh treatment just killing certain taxa? How acts DIC on skin microbiota directly or via the immune-system?

Answer from the authors: We thank the reviewer for this comment. We have now supplemented our discussion by a more detailed inclusion of studies investigating the pathogenic role of *S. aureus* and the less pathogenic role of CoNS in the context of skin malignancies. The decrease in relative and absolute *S. aureus* abundance which occurs after treatment completion, with median and mean reduction of *S. aureus* abundance and its association with treatment response, supports other reports, claiming *S. aureus* as important bacterial species with pro-tumorigenic potential. However, although most studies focussed on *S. aureus* as most important species of the *Staphylococcus* genus, there is also literature describing *S. epidermidis*, the most common species of CoNS, linked to squamous skin cancer progression in transplant patients (Krueger A, et al. 2022. Changes in the skin microbiome associated with squamous cell carcinoma in transplant recipients. ISME Communications 2:13.) Recent reports suggest that pathology may be context dependent and indicate that pathogens like *S. aureus* may asymptotically colonize the skin but mutualists like *S. epidermidis* or corynebacteria may then promote disease. We have discussed these aspects in lines 307-328 of the revised version of the manuscript.

Reviewer #2 (Comments for the Author):

In this manuscript, the use of diclofenac (DIC) and cold atmospheric plasma (CAP) on the skin microbiome in the treatment of actinic keratosis was studied.

The following comments are made:

1. Line 36. "tuff" is a gene? If so, put it.

Answer from the authors: We have now supplemented this statement as suggested.

2. Lines 86, 245-246. How did the treatments with CAP and DIC? You don't say it. It is a fundamental part of your work, there should be a section to explain the treatments.

Answer from the authors: We have now explained the treatment performed by CAP and DIC in the Materials and methods section, lines 354-360: „The ACTICAP study design included a washout phase of potentially prohibited medication, including photosensitizing agents, immunomodulators, antineoplastic agents and systemic retinoids for 4-12 weeks before treatment with CAP or DIC was started, depending on drug type. Patients were included after a minimum of 6 months after physical procedures (e.g. surgery, laser, PDT). The identified treatment areas were computationally randomized to either CAP that was applied for 180 seconds twice a week or topical DIC (diclofenac 3%) twice a day (8).”

3. Line 90. What extremities are you talking about? explain it

Answer from the authors: We have now specified the extremities in the revised version of the manuscript. Of 3 patients with AK of the extremities, two patients were treated with AK of the hands, and one patient with AK of the lower leg. We have also included this information in the Materials and Methods section, lines 363-364.

4. Line 100. What does "LefSe" mean? put it

Answer from the authors: We have now explained the LefSe (Linear discriminant effect size) analysis in the manuscript (line 125).

5. Line 132-133. "The reduction of the relative abundance of *Staphylococcus* of 16S rRNA sequences was due largely to CoNS-reduction". It is not clear why you say they are CoNS. How did you measure and verify it? explain it.

Answer from the authors: *S. aureus* is an important human pathogen of the genus *Staphylococcus* to cause various infectious diseases and is characterized by positive coagulase reaction. In contrast, *S. aureus* is distinguished from the heterogeneous group of coagulase-negative staphylococci (CoNS), which has been considered as less- or non-pathogenic and which is characterized by a negative coagulase reaction {Becker, 2014 #230}. As we used a *S. aureus*-specific taqman probe and a *Staphylococcus* genus specific probe, we calculated the relative CoNS abundance by the following equation:

rel. CoNS abundance = rel. *Staphylococcus* abundance – relative *S. aureus* abundance

We have included the calculation in the Material and methods section of the manuscript (lines 446-447).

6. Line 150. In line 142-143, it is well established that it is an R, but in line 150 nR appears and you do not say what it means or what its definition is. Correct it and define what is an nR.

Answer from the authors: We have now explained the abbreviation (line176).

7. Line 163. "NR", should be nR

Answer from the authors: We corrected this issue now.

8. Line 179-183. What interpretation do you give to these results, discuss it.

Answer from the authors:

The differences, associated with the location of the AK lesions, mimic those described for these skin microbiome regions in healthy individuals, indicating that the diseased skin microbiome is not basically altered and the regional microbiota characteristics are maintained in AK areas. We have supplemented this information in the revised version of the manuscript, lines 256-258.

9. Line 194. Capitalizing Gram is a proper name.

Answer from the authors: We corrected this issue.

10. You do not discuss the effect of CAP on the microorganisms found and their differences. Discuss the mechanism of CAP on bacteria and why the differences found.

Answer from the authors:

We thank the reviewer for his comment. Different mechanisms have been reported by which CAP may affect microorganisms. We have now discussed the known effects of CAP on microorganisms in the manuscript, lines 310-328: „Several mechanisms have been reported for antibacterial CAP activity. Reactive oxygen and nitrogen species produced by CAP can trigger programmed cell death with hallmarks of apoptosis by short-term treatment but long-term CAP treatment can physically destruct intracellular components (18). In addition, CAP may promote the potency of macrophages to kill *S. aureus* by improving the formation of degradative phagosomes (19). Besides the anti-inflammatory effects by inhibiting the prostaglandin synthesis, DIC was suggested to affect *in vitro* bacterial growth by modulating DNA synthesis (20). Despite the different nature of both treatments, the differences on the skin microbiome composition linked to the treatments in our study were minor. However, absolute abundance of *Staphylococcus* and CoNS were significantly lower at week 24 than at week 0 for DIC but not CAP. Also the reduction of absolute *S. aureus* abundance at week 36 was significant for DIC but not CAP. Patients classified as nR 12 weeks after end of CAP therapy were characterized by significantly higher *S. aureus* relative and absolute abundance than R, indicating that a high *S. aureus* abundance may be a biomarker for CAP treatment failure. Although most studies have focussed on *S. aureus* as most important species of the *Staphylococcus* genus, a recent report links *S. epidermidis*, a common species of CoNS, to SCC progression in transplant patients (14). Pathologies might be context dependent. Pathogens like *S. aureus* may asymptotically colonize the skin, whereas mutualists like *S. epidermidis* or corynebacteria sometimes then promote disease (17, 21, 22).“

11. Lines 306-307. What control strains of *S. aureus* and *S. epidermidis* did you use?

Answer from the authors: We have now specified the strains in this paragraph. We used the reference strains *S. aureus* DSMZ 799 and *S. epidermidis* (DSMZ 1498) (lines 430-433).

12. Line 387 and Figure 5. *Staphylococcus aureus* abbreviated as *S. aureus*

Answer from the authors: We have modified the abbreviations as suggested.

March 23, 2023

Dr. Jan Kehrmann
Universität Duisburg-Essen
Institute of Medical Microbiology
Virchowstr. 179
Essen 45147
Germany

Re: Spectrum04401-22R1 (Reduced *Staphylococcus* abundance characterizes the lesional microbiome of actinic keratosis patients after field-directed therapies)

Dear Dr. Jan Kehrmann:

Your manuscript has been accepted, and I am forwarding it to the ASM Journals Department for publication. You will be notified when your proofs are ready to be viewed.

Sincerely,

Victor Gonzalez
Editor, Microbiology Spectrum
